# Submarine Groundwater Discharge in a Coastal Bay: Evidence from Radon Investigations

**Manhua Luo** [1,2], **Yan Zhang** [1,2,*], **Hailong Li** [1,2,3], **Xuejing Wang** [3] and **Kai Xiao** [3]

1   MOE Key Laboratory of Groundwater Circulation and Environment Evolution and School of Water Resources and Environment, China University of Geosciences (Beijing), Beijing 100083, China; manhualuo2015@163.com (M.L.); hailongli@cugb.edu.cn (H.L.)
2   State Key Laboratory of Biogeology and Environmental Geology, China University of Geosciences (Beijing), Beijing 100083, China
3   Guangdong Provincial Key Laboratory of Soil and Groundwater Pollution Control and School of Environmental Science and Engineering, Southern University of Science and Technology, Shenzhen 518055, China; wangxj3@sustech.edu.cn (X.W.); xiaok@sustech.edu.cn (K.X.)
*   Correspondence: yanzhang@cugb.edu.cn

**Abstract:** Jiaozhou Bay, an urbanized coastal bay located in the southern part of Shandong Peninsula, China, has been deeply affected by anthropogenic activities. Here, the naturally occurring $^{222}$Rn isotope was used as a tracer to assess the submarine groundwater discharge (SGD) in this bay. The time series of $^{222}$Rn concentrations in nearshore seawater were monitored continuously over several tidal cycles at two fixed sites (Tuandao (TD) and Hongdao (HD)) during the dry season in spring and the wet season in autumn of 2016. $^{222}$Rn concentrations in seawater were negatively related to the water depth, indicating the influence of tidal pumping. A $^{222}$Rn mass balance model revealed that the mean SGD rates were 21.9 cm/d at TD and 17.8 cm/d at HD in the dry season, and were 19.5 cm/d at TD and 26.9 cm/d at HD in the wet season. These rates were about 8–14 times the discharge rates of the local rivers. Enhanced groundwater inputs occurred at HD in the wet season, likely due to the large tidal amplitudes and the rapid response to local precipitation. Large inputs of SGD may have important influences on nutrients levels and structure, as well as the water eutrophication occurring in coastal waters.

**Keywords:** submarine groundwater discharge; $^{222}$Rn; mass balance model; Jiaozhou Bay

## 1. Introduction

Submarine groundwater discharge (SGD) is defined as the flow of all continental margin waters from the seabed to the coastal ocean, which includes submarine fresh groundwater discharge (SFGD) and recirculated saline groundwater discharge (RSGD) [1,2]. As an important process of land–ocean interactions, SGD has received increasing attention in recent decades in the field of hydrogeology. Accumulating evidence has shown that SGD flux can be comparable to or even higher than riverine flux [2–5]. On a global scale, SGD flux is 3 to 4 times greater than the freshwater fluxes into the oceans by rivers [6]. Moreover, SGD contains high levels of nutrients [7–9], carbon [10,11], and metals [12,13]. As a result, a small SGD flux may carry a large quantity of dissolved materials from land to ocean [14,15]. These large fluxes of dissolved materials can lead to water eutrophication and acidification in coastal areas [16–18], which seriously influence marine ecosystems [19,20]. Therefore, it is of great significance to correctly understand and evaluate the SGD flux for coastal environment protection.

Approaches to evaluate SGD mainly include seepage meters [21–23], hydrological models [24–26], and geochemical tracers [27,28]. In particular, naturally occurring geochemical tracers, such as radium

($^{223,224,226,228}$Ra) and radon ($^{222}$Rn) isotopes, have been widely used in tracing SGD and water mixing processes in coastal systems [14,29–32]. $^{222}$Rn with a half-life of 3.8 days is chemically conservative and enters groundwater by the alpha recoil of their parent radionuclides in sediments [33,34]. Also, high resolution and automated $^{222}$Rn observations are easily performed, which makes it very useful for quantifying groundwater discharge over extended time scales.

A heavily urbanized coastal bay (Jiaozhou Bay, China) has been greatly influenced by anthropogenic activities such as rapid population growth and industrial expansion. From 1966 to 2008, an area of 180 km$^2$ of land has been reclaimed from the coastal regions by backfilling along the northern and western coastline of Jiaozhou Bay [35]. The nutrient concentrations (dissolved inorganic nitrogen (DIN), dissolved inorganic phosphorus (DIP)) in Jiaozhou Bay were increasing from the 1960s to the 1990s [36,37]. The bay waters underwent seasonal eutrophication and red tides occurred frequently because of large discharges of pollutants into the bay [38,39]. However, previous studies mainly have focused on regional SGD and associated nutrient fluxes in Jiaozhou Bay [40,41]. Little information is available on the temporal variations of SGD in this region. Therefore, it is of great significance to study the variability of SGD with different seasons and sites in Jiaozhou Bay.

In this study, the variations of SGD were assessed for Jiaozhou Bay over several tidal cycles at two fixed sites (Tuandao and Hongdao) in spring and autumn, based on continuous $^{222}$Rn observations. The difference in SGD between different seasons and sites was analyzed, and the relationship between the SGD and multiple influence factors, including water depth, season, location, and local river discharge, were explored.

## 2. Materials and Methods

### 2.1. Study Site

Jiaozhou Bay, a typical semiclosed bay (35°58′ N~36°18′ N, 120°04′ E~120°23′ E), is located in the southern part of Shandong Peninsula, China (Figure 1), linking to the Yellow Sea through a narrow channel 3.1 km wide. The average water depth is 7 m, and the bay mouth has a maximum water column depth of 60~70 m [42]. The average tidal range is 2.71 m and the largest tidal range is 6.87 m [43]. Tidal prism in 2008 decreased by 28% compared to the value in 1935 and the water residence time increased from 36 to 41 days due to land reclamation [44]. The water area had been reduced to 300 km$^2$ by 2015, due to the influence of human activities, and tidal flats occupy approximately one-third of the total water area [45–47].

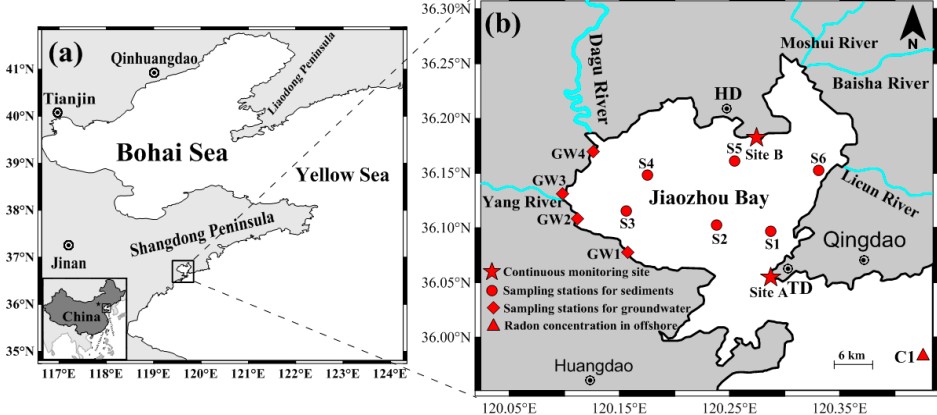

**Figure 1.** (**a**) Location of Jiaozhou Bay, China, and (**b**) the layout information of the radon sampling stations. TD: Tuandao, HD: Hongdao.

There are several different types of terrains surrounding the bay: Jimo Basin in the north, an alluvial plain in the northwest, Laoshan Mountain in the east, and Xiaozhu Mountain in the southwest [48]. The west and north of the bay have Quaternary unconsolidated sediments, which can easily store

groundwater and be recharged by rainfall. Figure 2 shows the daily precipitation and evaporation in 2016 with an annual mean rainfall of ~484 mm. The mean precipitation in spring 2016 and autumn 2016 (during the sampling period for this study) was 0.95 mm/d and 1.33 mm/d, and the mean evaporation was 3.37 mm/d and 2.62 mm/d, respectively. Many seasonal rivers were flowing into Jiaozhou Bay, including Yanghe River, Moshui River, Licun River, Dagu River, and Baisha River [42,49]. The Dagu River contributes at least 80% of the freshwater from the local rivers each year, with an average discharge rate of $1.98 \times 10^6$ m$^3$/d [50,51].

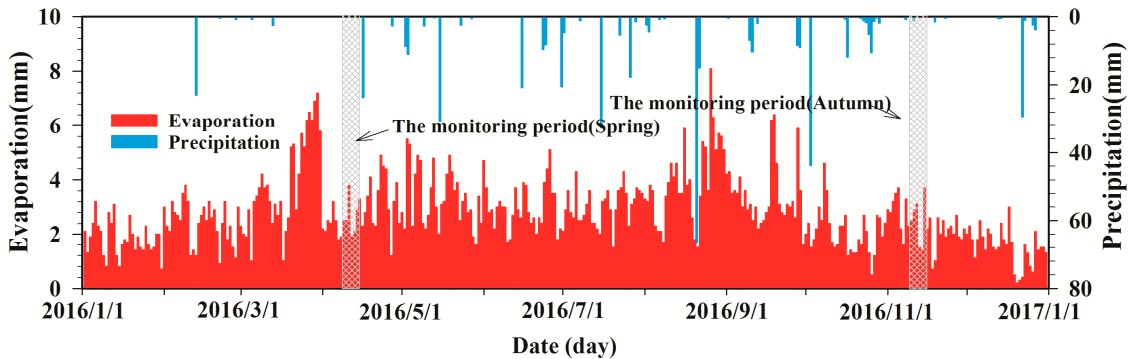

**Figure 2.** Daily precipitation and evaporation at the study sites in 2016. The gray area shows the monitoring period. Data are from the China Meteorological Data Network.

*2.2. Time-Series Deployments*

Continuous $^{222}$Rn monitoring was conducted at two fixed sites during the year's contrasting hydrological conditions (dry season in spring and wet season in fall). The first test site in Tuandao (TD) was located in the south, near the mouth of the bay. The second site in Hongdao (HD) was located in the north and surrounded by a more enclosed section of the bay, compared to TD (Figure 1). The two stations were separated by approximately 14.4 km. The time-series measurements of $^{222}$Rn concentrations were performed using a radon-in-air monitor and RAD AQUA [52]. Radon counts were integrated over 1 h cycles, giving measurement uncertainties of 10 ~ 30% for each data point. In the dry season, continuous measurements of $^{222}$Rn concentrations in the nearshore seawater were conducted from 9 to 11 April 2016 (49 h) at TD, and from 11 to 14 April 2016 (68 h) at HD. In the wet season, $^{222}$Rn concentrations were monitored from 10 to 12 November 2016 (50 h) at TD, and from 13 to 15 November 2016 (47 h) at HD.

During each time-series deployment, seawater was continuously pumped from ~0.5 m above the sea bottom and filtered through a cartridge filter to screen out any particulates before entering the AQUA system. The air–water exchanger was connected to the RAD7 [52] detector by a closed air loop. To ensure the accuracy of measurement, no bubble was introduced into the system. The $^{222}$Rn concentration in seawater was determined by multiplying the $^{222}$Rn concentration in the water column and the partition coefficient ($\alpha$). Schubert et al. introduced an easily applied equation to derive the partition coefficient [53], which depends on both water temperature and salinity, as shown in Equations (1)–(3):

$$A(^{222}Rn) = A(^{222}Rn_m) \times \alpha \tag{1}$$

$$\alpha = \frac{\beta T}{273.15} \tag{2}$$

$$\ln \beta = a_1 + a_2\left(\frac{100}{T}\right) + a_3 \ln\left(\frac{T}{100}\right) + S\left(b_1 + b_2\left(\frac{T}{100}\right) + b_3\left(\frac{T}{100}\right)^2\right) \tag{3}$$

where $A(^{222}Rn)$ and $A(^{222}Rn_m)$ are the $^{222}$Rn concentrations (Bq/m$^3$) before and after calibration, respectively; $\alpha$ is the partition coefficient; $\beta$ is the Bunsen coefficient; $T$ is the temperature (°C); $S$ is salinity (ppt); and $a_1$–$a_3$ and $b_1$–$b_3$ are six variable parameters.

In parallel with continuous monitoring of [222]Rn, the electrical conductivity, water temperature, and water levels were measured automatically in 30 min intervals by a CTD-Diver. The barometric pressure and temperature were recorded by a Baro-Diver placed in the air near the continuous [222]Rn monitoring sites. Continuous [222]Rn concentration in the atmosphere was also measured by an additional RAD7 detector. Both RAD7 detectors were programmed to integrate the [222]Rn counts every hour, whether in the air or the water. Wind speed was monitored by a portable hand-held anemometer over a 30 min period.

## 2.3. [226]Ra Enrichment and Analysis

To estimate the [222]Rn supported by [226]Ra in seawater, 60 L of seawater was collected from the [222]Rn monitoring sites to extract [226]Ra. After filtering, the seawater was slowly passed through a polyamide fiber, containing about 25 g of Mn fibers at a flow rate of less than 1 L/min. The fibers were thoroughly washed with distilled water to remove all particles and salts, and then the fibers were taken to the laboratory for measurement. In the laboratory, the fiber samples were sealed for 22 days (7 times the half-life of [222]Rn). After [226]Ra reached its decay equilibrium with [222]Rn and its daughter, the [226]Ra concentrations were analyzed with a RAD7 detector, following the method of Kim et al. [54]. To improve the accuracy, the measurement time of each sample was increased to 12 h. The measurement uncertainty of [226]Ra was 10–20%.

## 2.4. Determination of [222]Rn in Pore Water

Six sediment samples from Jiaozhou Bay were collected to evaluate the [222]Rn concentration in pore water based on sediment equilibration experiments [55] (Figure 1). In the laboratory, 100 g sediment samples were mixed with 500 mL of seawater in radon-free Erlenmeyer flasks, and then sealed and oscillated continuously for about 30 days. Given the half-life of [222]Rn, the [226]Ra and [222]Rn in the pore water of sediments and the overlying water reach decay equilibrium after ~4 weeks. The overlying water was then transferred to a 250 mL sampling bottle and measured using a RAD7 detector with a RAD-$H_2O$ accessory [52].

## 2.5. Radon Mass Balance Model

Burnett et al. developed a radon mass balance model based on continuous [222]Rn observations [31]. In seawater systems, the [222]Rn sources included SGD input, sediment diffusion, [226]Ra decay, and tidal input. Sinks of [222]Rn included tidal output, atmospheric evasion, decay loss, and mixing loss. The radon mass balance is shown in Equation (4):

$$\frac{dI}{dI} = F_{SGD} + F_{tide} + F_{sed} - F_{atm} - F_{mix} - F_{dec} \tag{4}$$

where $dI/dt$ is the difference in flux of [222]Rn between two successive measurements (1 h in this study) (Bq/m$^2$ h); $F_{SGD}$ is the [222]Rn flux attributed to SGD (Bq/m$^2$ h); $F_{tide}$ is the flux of [222]Rn induced by tidal input and output (Bq/m$^2$ h); $F_{sed}$ is the [222]Rn diffusion flux from sediments (Bq/m$^2$ h); $F_{atm}$ is the [222]Rn flux of atmospheric evasion (Bq/m$^2$ h); $F_{dec}$ is the decay loss flux of [222]Rn (Bq/m$^2$ h); and $F_{mix}$ is the [222]Rn flux out of the system caused by mixing (Bq/m$^2$ h). When the non-SGD source and sink terms are determined in Equation (4), the [222]Rn flux via SGD can be determined. Finally, the [222]Rn flux via SGD is easily converted to the SGD flux by dividing the [222]Rn concentration in groundwater end-member.

Note that the radon mass balance model alone did not allow a separation of these two SGD components of SFGD and RSGD [56]. Therefore, a simple two end-member mixing model on salinity was used to estimate the fresh groundwater fraction for the two seasons [57,58]:

$$S_{GW} = (1 - f_{fw}) \times S_{bay} + f_{fw} \times S_{fw} \tag{5}$$

where $S_{GW}$ is the groundwater salinity in the mixing zone; $S_{fw}$ and $S_{bay}$ are the average salinities of fresh groundwater and bay water, respectively; and $f_{fw}$ is the fraction of SFGD.

## 3. Results

### 3.1. $^{222}$Rn Inventory and $^{226}$Ra Supply

In the dry season, the $^{222}$Rn concentrations during the monitoring period ranged from 17.0 to 136.7 Bq/m$^3$ (average ± standard deviation: 57.8 ± 30.0 Bq/m$^3$) at TD and from 9.7 to 175.5 Bq/m$^3$ (77.4 ± 30.7 Bq/m$^3$) at HD. In the wet season, the $^{222}$Rn concentration ranged from 13.1 to 68.4 Bq/m$^3$ (31.0 ± 11.4 Bq/m$^3$) at TD, and from 41.9 to 293.8 Bq/m$^3$ (111.9 ± 48.4 Bq/m$^3$) at HD (Figure 3). The average tidal range was larger in the dry season (4.28 m) than that in the wet season (2.86 m) at TD. On the contrary, there was a lower tidal range in the dry season (3.31 m) than in the wet season (4.18 m) at HD. As shown in Figure 3, the concentration of $^{222}$Rn changed periodically with the tides and had a negative correlation with water depth. This indicated that low and high concentration of $^{222}$Rn in seawater was occurred during flood tide and ebb tide, respectively. The general fluctuations of $^{222}$Rn concentration during the monitoring period were known to respond to tidal pumping and the hydraulic gradient. Most likely, the hydraulic gradient between seawater and groundwater increased at low tides, which caused large amounts of seepage and higher $^{222}$Rn concentrations. Conversely, the hydraulic gradient decreased at high tides. The $^{222}$Rn concentration in the nearshore water was diluted by mixing with offshore water at high tides, which contributed to less seepage and lower $^{222}$Rn concentrations. Interestingly, there was a lag time between low tide and the peak of $^{222}$Rn concentration in this study, and the lag times were different at different low tides (Figure 3). This phenomenon was consistent with previous studies [9,59]. Wu et al. showed that the phase delay may be associated with the distance from the shoreline, water depth, and topographic conditions [59].

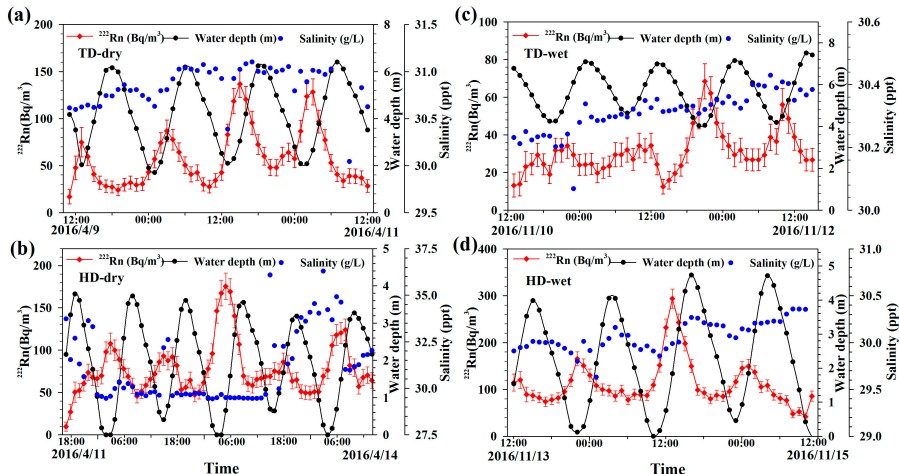

**Figure 3.** Continuous variation of $^{222}$Rn concentration and salinity with the tides at (**a**) TD–dry season, (**b**) HD–dry season, (**c**) TD–wet season, and (**d**) HD–wet season.

As shown in Figure 3a,d, the salinity has a significantly positive correlation with water depth, indicating that the salinity peaked at high tides, and the variation of salinity tended to increase with time, which may be due to an increased inflow of high salinity seawater from the open ocean, accompanied by a gradual increase in tidal range. The salinity fluctuation (29.5–36.3 ppt) at HD in the dry season (Figure 3b) was the largest (Figure 3).

Figure 4 shows the temporal variation of $^{226}$Ra in seawater at both sites in two seasons. In the dry season, the $^{226}$Ra concentration in seawater ranged from 3.3 to 4.4 Bq/m$^3$ (3.8 ± 0.3 Bq/m$^3$) at TD, and from 2.2 to 5.8 Bq/m$^3$ (3.5 ± 1.0 Bq/m$^3$) at HD. In the wet season, $^{226}$Ra concentration ranged from 2.1 to 8.0 Bq/m$^3$ (4.1 ± 2.2 Bq/m$^3$) at TD, and from 1.9 to 8.2 Bq/m$^3$ (3.7 ± 1.8 Bq/m$^3$) at HD. The $^{226}$Ra concentration in seawater did not change significantly during the measurement period. Compared with the $^{222}$Rn in seawater, $^{226}$Ra had a lower concentration, showing that the inputs of radon from the ingrowth of $^{226}$Ra were limited.

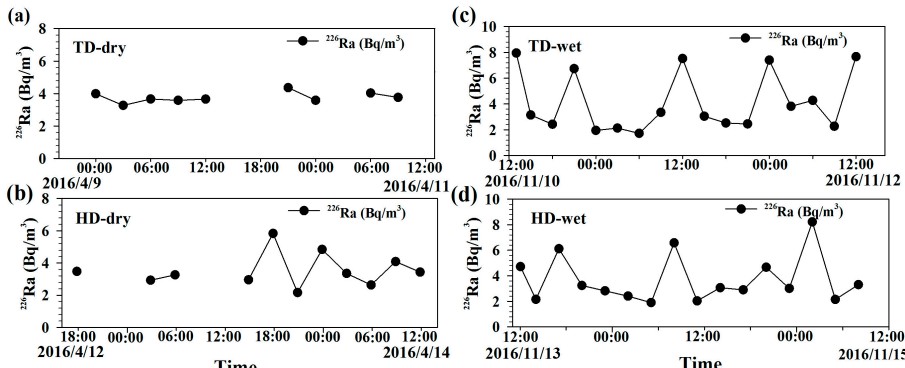

**Figure 4.** Temporal variation of $^{226}$Ra in seawater at (**a**) TD–dry season, (**b**) HD–dry season, (**c**) TD–wet season, and (**d**) HD–wet season.

To evaluate the $^{222}$Rn flux attributed to SGD, it is necessary to calculate the variations in storage flux of $^{222}$Rn ($dI/dt$) in seawater between two adjacent time intervals. The excess $^{222}$Rn inventory was defined as the product of the excess $^{222}$Rn concentration (subtracting $^{226}$Ra from total $^{222}$Rn) and water depth [31]:

$$I(t) = \left[ ^{222}Rn(t) - ^{226}Ra(t) \right] \times h(t) \tag{6}$$

where *I* is the $^{222}$Rn inventory (Bq/m$^2$); $^{222}Rn(t)$ and $^{226}Ra(t)$ are the average concentrations of $^{222}$Rn and $^{226}$Ra for each time interval, respectively (Bq/m$^3$); and $h(t)$ is water depth (m) at time *t*. The variation of excess $^{222}$Rn inventory ($dI/dt$) can be estimated by the difference of $^{222}$Rn inventory between two adjacent time intervals. Negative values of $dI/dt$ indicate a decrease in $^{222}$Rn storage, while positive values show an increase in $^{222}$Rn storage. In the dry season, $dI/dt$ ranged from −97.0 to 140.9 Bq/m$^2$ h at TD, and from −122.2 to 149.2 Bq/m$^2$ h at HD. In the wet season, $dI/dt$ ranged from −83.2 to 86.0 Bq/m$^2$ h at TD, and from −275.0 to 362.0 Bq/m$^2$ h at HD.

### 3.2. Tidal Effects

Radon inventories in seawater can be strongly controlled by tides. $^{222}$Rn from the open sea can flow into the seawater system with the incoming water during the flood tide and would flow out of the seawater system with the outgoing water during the ebb tide. Following Zhang et al. [9], the flux of $^{222}$Rn ($F_{tide}$) subjected to tide transportation can be expressed using Equation (7):

$$F_{tide} = \begin{cases} \frac{h_{t+\Delta t} - h_t}{\Delta t} \times \left[ b^{222}Rn_{mean} + (1-b)^{222}Rn_{off} \right] & \text{when flood tide} \\ \frac{h_{t+\Delta t} - h_t}{\Delta t} \times ^{222}Rn_{t,t+\Delta t} & \text{when ebb tide} \end{cases} \tag{7}$$

where $h_{t+\Delta t}$ and $h_t$ are the water depth (m) at time *t* and $t+\Delta t$, respectively; $^{222}Rn_{mean}$ and $^{222}Rn_{off}$ are the mean concentration of $^{222}$Rn in the seawater column and in offshore water (Bq/m$^3$), respectively; *b* is the return flow factor (percentage of the tidal prism that returns from the open sea during a rising tide); and $^{222}Rn_{t,t+\Delta t}$ is the average concentration of $^{222}$Rn (Bq/m$^3$) over a time interval. When $h_{t+\Delta t} > h_t$, $F_{tide}$ equals to the flux of $^{222}$Rn entering with the incoming tide ($F_{tide-Influx}$), while $h_{t+\Delta t} < h_t$, $F_{tide}$ equals to the flux of $^{222}$Rn leaving with the outgoing tide ($F_{tide-Outflux}$), based on the tidal prism model [33,60], which considered the difference with the water volumes between high tide and low tide in different study areas of the bay. The return flow factor *b* was calculated to be 0.89 and 0.92 in the dry and wet seasons, respectively. The $^{222}$Rn concentrations for both sites in offshore water (station C1 in Figure 1) were 20.5 Bq/m$^3$ in the dry season and 13.1 Bq/m$^3$ in the wet season, respectively. According to Equation (7), the flux of $^{222}$Rn affected by tides ($F_{tide-Influx}$ and $F_{tide-Outflux}$) can be calculated. In the dry season, $F_{tide-Influx}$ ranged from 6.1 to 67.6 Bq/m$^2$h at TD, and from 7.2 to 84.3 Bq/m$^2$h at HD, $F_{tide-Outflux}$ ranged from −90.7 to −0.4 Bq/m$^2$ h at TD, and from −63.4 to −3.5 Bq/m$^2$ h at HD. In the wet season, $F_{tide-Influx}$ ranged from 0.2 to 26.1 Bq/m$^2$ h at TD, and from 2.0 to 136.0 Bq/m$^2$ h at HD, $F_{tide-Outflux}$ ranged from −21.6 to −0.8 Bq/m$^2$ h at TD, and from −90.1 to −0.4 Bq/m$^2$ h at HD.

### 3.3. Atmospheric Loss

Radon is a slightly soluble gas in water. When the $^{222}$Rn concentration in water is greater than that in air, $^{222}$Rn will diffuse into the atmosphere through the water–air interface. The atmospheric loss of $^{222}$Rn was mainly related to its migration coefficient and the concentration gradient of the water–gas interface [61]:

$$F_{atm} = k(C_w - \alpha C_{air}) \tag{8}$$

where $k$ is the gas transfer velocity (m/h); $C_w$ and $C_{air}$ are the $^{222}$Rn concentrations in seawater and atmosphere, respectively (Bq/m$^3$). $\alpha$ is the partition coefficient related to temperature and salinity (see Equation (2)). When $C_w > \alpha C_{air}$, $^{222}$Rn diffuses from the water column to the air. Lambert et al. give the expression of the gas transfer velocity $k$ of $^{222}$Rn [62]:

$$k = \begin{cases} 0.45\mu^{1.6}(S_c/600)^{-0.5}, & \mu > 3.6 m/s \\ 0.45\mu^{1.6}(S_c/600)^{-0.6667}, & 1.5 \le \mu \le 3.6 m/s \\ 0.91, & \mu < 1.5 m/s \end{cases} \tag{9}$$

$$S_c = 3417.6e^{-0.0634 \times T} \tag{10}$$

where $\mu$ is the wind speed (m/s) and $S_c$ is Schmidt constant for radon at a given water temperature [63]. Figure 5 shows the temporal variations of $^{222}$Rn in the air with the observed wind speed in both seasons. In the dry season, the $^{222}$Rn concentration in the air ranged from 0.4 to 19.4 Bq/m$^3$ (4.93 ± 3.9 Bq/m$^3$) at TD, and from 0 to 14.2 Bq/m$^3$ (2.86 ± 2.7 Bq/m$^3$) at HD. The wind speed ranged from 0 to 6.9 m/s (3.2 ± 2.3 m/s) at TD, and from 0 to 8.7 m/s (4.9 ± 2.3 m/s) at HD. In the wet season, the $^{222}$Rn concentration in the air ranged from 0 to 21.3 Bq/m$^3$ (8.70 ± 4.6 Bq/m$^3$) at TD, and from 1.4 to 24.2 Bq/m$^3$ (9.32 ± 5.7 Bq/m$^3$) at HD. The wind speed ranged from 0.2 to 5.5 m/s (1.7 ± 1.4 m/s) at TD, and from 0 to 6.1 m/s (2.0 ± 1.5 m/s) at HD (Figure 5).

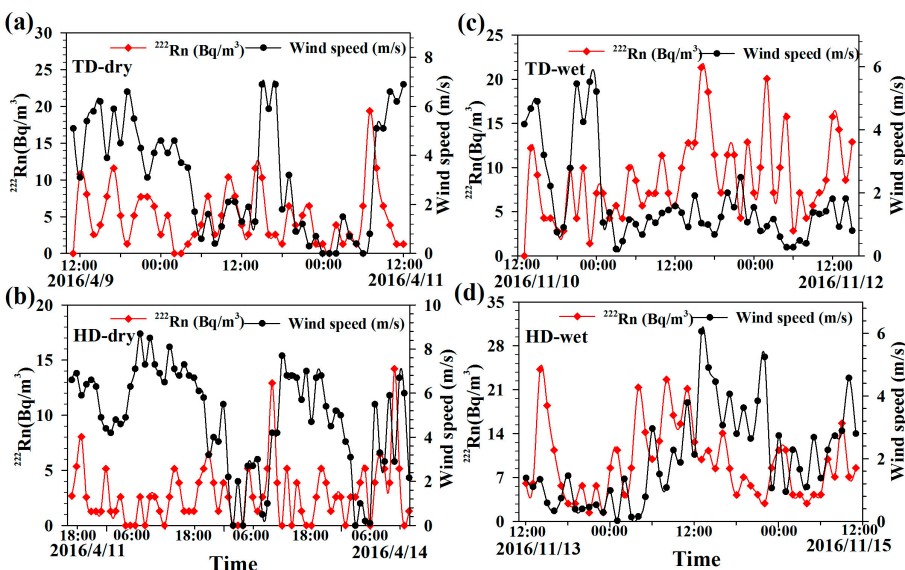

**Figure 5.** Temporal variations of $^{222}$Rn concentrations in air and the wind speed at (**a**) TD–dry season, (**b**) HD–dry season, (**c**) TD–wet season, and (**d**) HD–wet season.

Based on Equations (8–10), the diffusive flux of $^{222}$Rn ($F_{atm}$) across the water–air interface was determined (Figure 6). The negative and positive values of $F_{atm}$ represent the net decrease and increase of atmospheric loss of $^{222}$Rn in the adjacent period, respectively (Figure 6). In the dry season, this flux ranged from −5.7 to 8.2 Bq/m$^2$ h at TD, and from −3.7 to 5.0 Bq/m$^2$ h at HD. In the wet season, this flux ranged from −0.6 to 0.9 Bq/m$^2$ h at TD, and from −7.1 to 14.7 Bq/m$^2$ h at HD.

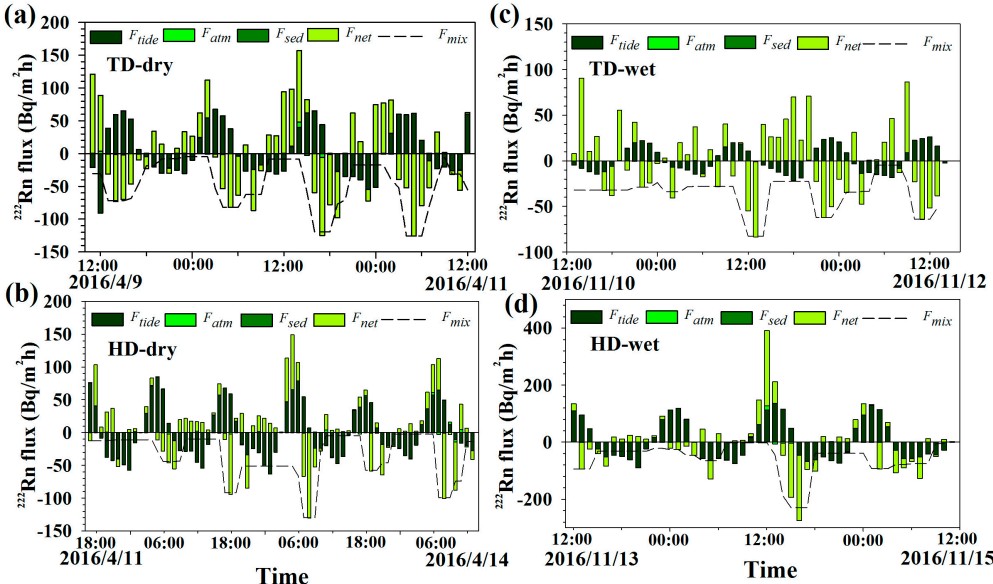

**Figure 6.** Contributions of $^{222}$Rn flux from the SGD, mixing, atmospheric loss, sediments, and tidal transportation at (**a**) TD–dry season, (**b**) HD–dry season, (**c**) TD–wet season, and (**d**) HD–wet season. The algebraic sum of all these fluxes is the excess $^{222}$Rn inventory variation. Note: $F_{net} = F_{SGD} - F_{mix}$.

### 3.4. Diffusive Flux From Bottom Sediments

Radon exchange between sediments and the overlying seawater was affected by the concentration gradient, as well as biological and physical disturbances. Sediment source of radon is important due to radon's short half-life. The $^{222}$Rn flux through sediment diffusion was obtained by performing a sediment cultivation experiment of radon in this study [64]:

$$F_{sed} = (\lambda n D_m)^{0.5}\left(C_{eq} - C_0\right) \tag{11}$$

where $F_{sed}$ is the diffusion flux of $^{222}$Rn from sediments (Bq/m$^2$ h); $\lambda$ is the decay constant of $^{222}$Rn (0.181d$^{-1}$); $D_m$ is the molecular diffusion coefficient related to temperature; and $C_{eq}$ is the $^{222}$Rn concentration (Bq/m$^3$) in the pore water of sediments. $C_0$ is the $^{222}$Rn concentration (Bq/m$^3$) in the overlying seawater. After correcting the wet density and porosity of the sediment, $C_{eq}$ and $D_m$ can be obtained by the following equations:

$$C_{eq} = \frac{C_{sed} \times \rho_{wet}}{n}, \ C_{sed} = \frac{C_0 \times V}{M_{sed}}, \ D_m = 10^{-(1.59 + 980/(T+273))} \tag{12}$$

where $C_{sed}$ is the $^{222}$Rn concentration in wet sediments (Bq/kg); $V$ is the volume of the overlying seawater in the sediment cultivation experiment; $\rho_{wet}$ is the wet density of the sediments (kg/m$^3$); and $M_{sed}$ is the mass of the sediments (kg). According to the laboratory particle analysis, the porosity of the sediments in Jiaozhou Bay was 0.3 and the wet density of the sediments was 1205 kg/m$^3$. The average $^{222}$Rn concentration in the pore water obtained from the sediment cultivation experiment was 4745 Bq/m$^3$ at TD (S1–S3 in Figure 1) and 4788 Bq/m$^3$ at HD (S4–S6 in Figure 1). In the dry season, the average value of the molecular diffusion coefficient ($D_m$) was $8.75 \times 10^{-6}$ cm$^2$/h at TD and $9.86 \times 10^{-6}$ cm$^2$/h at HD. Based on Equation (11), the variations in $^{222}$Rn flux across the sediment–water interface ranged from $-2.88 \times 10^{-2}$ to $3.46 \times 10^{-2}$ Bq/m$^2$ h at TD and from $-6.78 \times 10^{-2}$ to $2.88 \times 10^{-2}$ Bq/m$^2$ h at HD. In the wet season, the average value of the molecular diffusion coefficient ($D_m$) was $1.03 \times 10^{-5}$ cm$^2$/h at TD and $9.70 \times 10^{-6}$ cm$^2$/h at HD. The variations in $^{222}$Rn flux across the sediment–water interface ranged from $-1.97 \times 10^{-2}$ to $2.39 \times 10^{-2}$ Bq/m$^2$ h at TD and from $-5.50 \times 10^{-2}$ to $3.75 \times 10^{-2}$ Bq/m$^2$ h at

HD. Compared to the flux of $^{222}$Rn from tides, the diffusion flux of $^{222}$Rn from sediments was only a small portion.

### 3.5. Mixing Loss and SGD Flux

After $dI/dt$ was corrected for atmospheric loss, tidal effects, and diffusion from sediments, the net $^{222}$Rn flux could be estimated, i.e., the algebraic sum of $F_{SGD}$ and $F_{mix}$. The radon decay loss in Equation (4) was not considered because the decay loss of $^{222}$Rn within 1 h was negligible. The net flux of $^{222}$Rn ($F_{net}$) can be expressed as:

$$F_{net} = F_{SGD} - F_{mix} \tag{13}$$

To avoid an overestimation of SGD, the maximum negative value of $F_{net}$ was chosen as the mixing loss [31,61]. As shown in Figure 6, in the dry season, the net $^{222}$Rn flux ranged from −125.6 to 120.9 Bq/m$^2$ h at TD, and from −129.8 to 83.3 Bq/m$^2$ h at HD. This means that the $^{222}$Rn flux of SGD varied from 0 to 151.4 Bq/m$^2$ h at TD, and from 0 to 147.5 Bq/m$^2$ h at HD. In the wet season, the net $^{222}$Rn flux ranged from −82.5 to 90.3 Bq/m$^2$ h at TD, and from −228.8 to 264.1 Bq/m$^2$ h at HD. Therefore, the $^{222}$Rn flux of SGD varied from 0 to 122.5 Bq/m$^2$ h at TD, and from 0 to 265.8 Bq/m$^2$ h at HD. Table 1 shows the $^{222}$Rn sources and sinks in the dry and wet seasons. The $^{222}$Rn fluxes of atmospheric loss represented only a small portion of the total $^{222}$Rn sources.

**Table 1.** The mean values and standard deviations of $^{222}$Rn sources and sinks in the dry and wet seasons at TD and HD.

| $^{222}$Rn Fluxes (Bq/(m$^2$ h)) | TD (Dry Season) | HD (Dry Season) | TD (Wet Season) | HD (Wet Season) |
|---|---|---|---|---|
| *Sinks* | | | | |
| $F_{atm}$ | 0.02 ± 1.65 | 0.05 ± 1.49 | $(-0.31 \pm 22.5) \times 10^{-2}$ | −0.02 ± 6.69 |
| $dI/dt$ | 0.65 ± 54.4 | 1.71 ± 52.2 | 2.13 ± 33.2 | −3.53 ± 103.6 |
| $F_{mix}$ | −47.0 ± 38.2 | −35.7 ± 35.1 | −35.9 ± 19.0 | −58.8 ± 56.7 |
| $F_{tide-Outflux}$ | −26.2 ± 17.6 | −28.1 ± 15.1 | −10.4 ± 11.8 | −50.4 ± 18.3 |
| *Sources* | | | | |
| $F_{tide-Influx}$ | 45.9 ± 18.5 | 45.5 ± 22.6 | 15.6 ± 7.5 | 80.0 ± 39.4 |
| $F_{sed}$ | $(0.01 \pm 1.26) \times 10^{-2}$ | $(-0.06 \pm 2.02) \times 10^{-2}$ | $(-0.10 \pm 8.41) \times 10^{-3}$ | $(0.02 \pm 1.99) \times 10^{-2}$ |
| $F_{SGD}$ | 44.5 ± 36.9 | 36.3 ± 36.7 | 37.0 ± 32.2 | 51.2 ± 54.8 |

The average $^{222}$Rn concentrations of groundwater at four stations (GW1–GW4 in Figure 1) were used to represent the groundwater end-member values at both sites. The $^{222}$Rn concentrations of groundwater ranged from 2850 to 7400 Bq/m$^3$ (4883 ± 1974 Bq/m$^3$) in the dry season, and from 2868 to 7960 Bq/m$^3$ (4563 ± 2246 Bq/m$^3$) in the wet season. The $^{222}$Rn flux attributed to the SGD should be divided by the $^{222}$Rn concentration in the groundwater end-member value entering the system. Figure 7 shows the variations in the SGD flux during the time-series measurements at both sites in two seasons. The SGD flux was negatively correlated with tidal height at TD in two seasons, which indicated that the mechanism of tidal pumping was an important process for driving SGD on a short time scale at TD in different seasons. Whereas, the SGD flux was positively correlated with tidal height at HD in different seasons (Figure 7). In the dry season, the SGD flux ranged from 0 to 74.4 cm/d (21.9 ± 18.3 cm/d) at TD and from 0 to 72.5 cm/d (17.8 ± 18.2 cm/d) at HD. In the wet season, the SGD flux varied from 0 to 64.5 cm/d (19.5 ± 17.1 cm/d) at TD, and from 0 to 139.8 cm/d (26.9 ± 29.2 cm/d) at HD. The SGD flux in the wet season at HD was the largest and about 1.5 times that of HD in the dry season. The SGD flux in the wet season at TD was about the same as in the dry season.

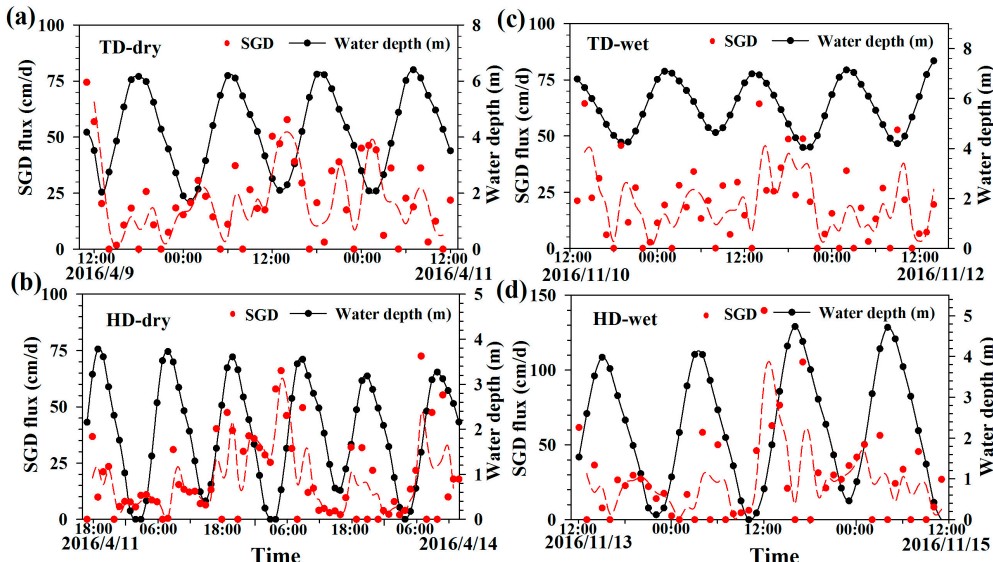

**Figure 7.** Temporal variations of SGD and water depth at (**a**) TD–dry season, (**b**) HD–dry season, (**c**) TD–wet season, and (**d**) HD–wet season. The dashed curves are moving averages corresponding to the obtained SGD flux.

According to Equation (4) and the data of field measurements, the $S_{GW}$, $S_{fw}$, and $S_{bay}$ were 30.6, 24.8, and 31.0 ppt in the dry season, respectively, and were 32.3, 29.5, and 33.1 ppt in the wet season, respectively. Therefore, the fresh groundwater fractions in the dry and wet seasons were 7% and 22%, respectively.

## 4. Discussion

### 4.1. Uncertainty Analysis

The estimate of SGD was influenced by multiple uncertain factors, among which the determination of groundwater end-member can make the biggest difference [9,14]. Other factors include the $^{222}$Rn concentrations from sediments, water level, and wind speed, which contribute to smaller SGD fluxes.

In this study, the $^{222}$Rn concentrations of the four groundwater samples were collected from different wetland types (sand beaches, mudflats, tidal marshes, and estuarine intertidal zones). These samples were used as the groundwater end-member value in the mass balance calculation. The mean values and standard deviations of $^{222}$Rn concentration of the groundwater samples were $4883 \pm 1974$ Bq/m$^3$ in the dry season and $4563 \pm 2246$ Bq/m$^3$ in the wet season. To evaluate the uncertainty in the SGD calculation caused by the groundwater end-member value, a variation in the mean groundwater end-member value of one sigma standard deviation ($\pm 40\%$ for the dry season and $\pm 49\%$ for the wet season) was considered in this study. Therefore, if the groundwater end-member value increases by 40% in the dry season or by 49% in the wet season, the mean SGD rate will decrease by 29% or 33%, respectively. If the groundwater end-member value decreases by 40% in the dry season or by 49% in the wet season, the mean SGD flux will increase by 67% or 96%, respectively.

### 4.2. SGD Fluxes for Different Sites and Seasons

In this study, the SGD rate showed significant differences between the TD and HD sites. The SGD rate in the dry season was larger at TD (21.9 cm/d) than at HD (17.8 cm/d), which was consistent with the tidal range at both sites. Whereas, the SGD flux in the wet season at HD was about 1.4 times that of TD, which was also consistent with the tidal amplitude at two sites. Moreover, the sediment characteristics at both sites are different. The strata around TD is the bedrock of Laoshan Mountain, which has a low permeability. However, the surface sediments in the Jimo Basin, around HD in the

northern part of the bay, consist of sand and silt with high permeabilities. This may affect the SGD flux at the two sites [14].

The SGD rates at the TD site did not show large seasonal variations because of the bedrock with low permeability around TD, which is unlikely to be sensitive to rainfall. Figure 7 shows that the SGD flux at TD in two seasons had a negative correlation with water depth during the monitoring period. Therefore, the SGD rate at TD was slightly larger in the dry season than in the wet season, due to large tidal amplitude. Whereas, the SGD rate at HD was larger in the wet season (26.9 cm/d) than in the dry season (17.8 cm/d), which was still consistent with the average tidal range in the two seasons. Figure 2 shows that the total precipitation in Jiaozhou Bay during the spring (March, April, and May 2016) and autumn (September, October, and November 2016) were 87.2 mm and 120.7 mm, respectively. Moreover, the rainfall in the month before the field monitoring periods in spring and autumn were 2.8 mm and 27.9 mm, respectively, which was consistent with the SGD flux at HD in the dry and wet seasons. In addition, the HD site is surrounded by surface sediments of sand and silt with high permeabilities. Also, the northern part of the bay has Quaternary unconsolidated sediments around HD that easily store groundwater, and the sediments were recharged by rainfall. Therefore, the SGD rate at HD was larger in the wet season than in the dry season, due to higher rainfall in the wet season that led to a greater hydraulic gradient between groundwater and seawater [51]. Collectively, the highest SGD rate for both sites was observed at HD in the wet season, likely due to the sedimentary lithology of sand and silt with high permeability, large tidal amplitude, and the rapid response to local precipitation.

*4.3. Comparisons with Local River Inputs*

The product of the SGD rate and the seepage area of Jiaozhou Bay determined the magnitude of the SGD flux into the bay [14]. Detailed information on the water area and tidal flats of the bay, and the average discharge rate of Dagu River was well described in Section 2.1. Tidal flats of sand and silt sediments with a large-scale seepage face mainly occur in the north and northwest of the bay due to a low slope and large tidal amplitudes [41]. To ensure the accuracy of the SGD flux estimation, the mean SGD rate (17.8 cm/d in the dry season and 26.9 cm/d in the wet season) at HD near tidal flats was used for estimating the magnitude of the SGD flux into the bay. Therefore, the SGD flux was about 8–14 times the freshwater inputs from the Dagu River.

*4.4. Comparisons with Previous Studies*

Most previous studies have focused on regional SGD in Jiaozhou Bay [40,41,65]. However, in this study, two fixed sites were selected for estimating the temporal variations of SGD over spatial and temporal scales. The influence of different seasons on the SGD was analyzed, both qualitatively and quantitatively.

In addition, several studies have assessed the SGD flux in Jiaozhou Bay using different methods. Yuan et al. showed that the SGD flux was 9 cm/d in the wet season (September and October 2011) and 4 cm/d in the dry season (April and May 2012) using the radium mass balance models ($^{224}$Ra and $^{226}$Ra) [65]. Qu et al. used a generalized Darcy's law to evaluate the SGD in four wetland types, including sandy beaches, mudflats, tidal marshes, and estuarine intertidal zones, with a range of $3.6 \times 10^{-3}$ to 7.6 cm/d [41]. Zhang et al. applied the $^{222}$Rn mass balance model to estimate the SGD rate (12.3 cm/d in January 2016 and 17.8 cm/d in July 2015) at TD, which was 25% lower than the results found in this study, likely due to the different seasons and the determinations of groundwater end-members [51]. However, compared with the SGD rates in other studies, it is suggested that the SGD in this study is in the range of those in previous studies (Table 2). The minor differences between the estimations may be a result of different study areas and different times.

**Table 2.** Comparison of SGD fluxes with previous studies in Jiaozhou Bay and other bay systems worldwide.

| Study Site | Methodology | SGD (cm/d) | References |
|---|---|---|---|
| Jiaozhou Bay, China | $^{224}$Ra and $^{226}$Ra | 9 (autumn), 4 (spring) | [65] |
| | $^{223}$Ra, $^{224}$Ra, $^{226}$Ra, $^{228}$Ra | 4.9~7.5, 4.3~6.5, 4.0~7.2, 3.0~5.7 | [40] |
| | Darcy's law | $3.6 \times 10^{-3}$~7.6 | [41] |
| | $^{222}$Rn | 12.3 (winter) 17.8 (summer) | [51] |
| | $^{222}$Rn | TD: 21.9 (dry season) 19.5 (wet season) HD: 17.8 (dry season) 26.9 (wet season) | This study |
| Pearl River Estuary, China | $^{224}$Ra | 23~50 (dry season) 6~14 (wet season) | [66] |
| Richmond River Estuary, Australia | $^{222}$Rn | 6~10 (dry season) 37~59 (wet season) | [67] |
| Maowei Sea, China | $^{222}$Rn | 3~69 (wet season) 2~38 (dry season) | [68] |
| Copano Bay, South Texas, USA | $^{222}$Rn | Reef and margin: 51.3~73.9 (spring) 17.9~39.5 (summer) Paleovalleys: 23~40.8 (spring) 12.3~26.7 (summer) | [69] |
| Jepara, Indonesia | $^{222}$Rn | 37 (Awur) 52 (Bandengan) | [70] |
| Daya Bay, China | $^{222}$Rn | 28.2 (northwest site) 30.9 (middle-east site) | [14] |

## 5. Conclusions

This study reported a point-scale evaluation of SGD into an urbanized bay (e.g., Jiaozhou Bay in China). A $^{222}$Rn mass balance model including tidal effects, diffusion from sediments, atmospheric loss, and mixing loss was applied to estimate the SGD flux over several tidal cycles in the dry and wet seasons.

$^{222}$Rn concentrations in seawater varied periodically with the tides and had a negative correlation with water depth during the monitoring period, suggesting the impact of tidal effects and the hydraulic gradient. Moreover, relatively high $^{222}$Rn concentrations were detected at HD during the wet season, implying enhanced groundwater inputs.

Based on the continuous monitoring of $^{222}$Rn concentration at two fixed sites, the mean SGD fluxes in the dry season were estimated to be 21.9 cm/d at TD and 17.8 cm/d at HD. The mean SGD rates in the wet season were 19.5 cm/d at TD and 26.9 cm/d at HD. The fresh groundwater fractions in the dry and wet seasons were 7% and 22%, respectively. The enhanced groundwater inputs occurred at HD in the wet season, which suggested that higher rainfall and the large tidal amplitude caused larger SGD fluxes. The SGD rates were about an order of magnitude greater than the discharges of the local rivers. This study highlighted the need to employ different analysis methods and sampling strategies to characterize the SGD fluxes occurring over spatial and temporal scales. Further investigations are needed to improve the understanding of the effects of SGD on marine environments in coastal systems.

**Author Contributions:** Conceptualization, M.L. and Y.Z.; methodology, M.L. and Y.Z.; validation, Y.Z.; formal analysis, M.L.; investigation, M.L., Y.Z., K.X., and X.W.; resources, Y.Z.; data curation, M.L. and Y.Z.; writing—original draft preparation, M.L.; writing—review and editing, Y.Z., H.L., K.X., and X.W.; supervision, Y.Z., H.L., K.X., and X.W.; project administration, Y.Z. and H.L.; funding acquisition, Y.Z. and H.L. All authors have read and agreed to the published version of the manuscript.

**Funding:** This research was funded by China Postdoctoral Science Foundation (Grant No. 2020M670399) and the National Natural Science Foundation of China (Grant Nos. 41972260, 41907162).

**Acknowledgments:** The authors thank our team members Ping Yuan, An An, Meng Zhang, Yanman Li, and Xiaoting Lu for their field assistance and laboratory measurements.

**Conflicts of Interest:** The authors declare no conflict of interest.

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
