# Peer review of "Submarine Groundwater Discharge in a Coastal Bay: Evidence from Radon Investigations"

_water, doi:10.3390/w12092552_

Round 1

Reviewer 1 Report

As the authors mention in sect. 3.2, radon inventories in the coastal sea can be strongly controlled by the tides. However, while the authors discuss offshore interaction (the tidal prism approach), they do not refer to the cyclic interaction between coastal water and subterranean estuary. SGD should generally be subdivided into fresh SGD and recirculated SGD (see references below). The authors should at least mention this in their discussion.

Reference:

Schubert, M., Petermann, E., Stollberg, R., Gebel, M., Scholten, J., Knöller, K., Lorz, C., Glück, F., Riemann, K., Weiss, H. (2019): Improved Approach for the Investigation of Submarine Groundwater Discharge by Means of Radon Mapping and Radon Mass Balancing, Water 11, 749; doi:10.3390/w11040749.

Reviewer 2 Report

Review of Submarine Groundwater Discharge in an urbanized coastal bay: evidence from radon investigations by Luo et al.

The article investigated the SGD fluxes at two different locations in Jiaozhou Bay, China. They performed a radon mass balance to estimate the SGD Rn flux, and used Rn concentrations in groundwater to convert the SGD Rn flux into a SGD water flux. Finally, they compared the SGD fluxes to river inputs and literature data.

Overall, the paper is well organized and follows a logical progression. However, I believe several modifications, some of them are major, can be done to improve the quality of the paper before publications. 

  1. I recommend to change the title of the paper. Indeed, the “urban” characteristic of the study area is not used. Based on the title, I was expecting to find results and discussion related with the anthropic activities of the area, such as fluxes of nutrients or pollutants. The quality and impact of the paper would be greatly improved if the authors combine the SGD fluxes to other parameters from the bay to illustrate the impact of these SGD fluxes as it is well known that SGD fluxes greatly impact the quality of the coastal waters. 
  2. In the introduction, the authors decided to mention fresh and marine SGD, but do not provide an estimate for these two components of the total SGD flux while they highlighted the importance of precipitation and hydraulic gradient. Could the author do a back of the envelope mass balance on salinity to try to tease apart the fresh SGD contribution ?
  3. Figure 1: add “TD” and “HD” to the map and in the legend insert that HD stands for Hondao and TD for Tuandao. This would greatly helps the reader.
  4. Figures 3, 4,5, 6 etc add TD wet, TD dry, HD wet, and HD dry as title of each panel. Again, this will help the reader when looking at the text and the figures together. 
  5. I was confused by the dry and wet seasons based on precipitation and evaporation data. Lines 83-88, we can see for April 2016 precipitation of 0.89 mm/d and evaporation of 2.86 mm/d so +0.89-2.86 = -1.97 mm/d. Similarly, in Nov 2016 there was +0.13 mm/d of precipitation and -2.37 mm/d of evaporation, which lead to -2.24 mm/d. However, the authors said that April (spring) was the dry season and November (fall) was the wet season, while the net deficit of rain - evaporation is larger in fall than in spring. 
  6. I am sure that some precautions were taken in the methods even if it was not described in the text. For example, when they filtered water through a cartridge filter before the AQUA system, they need to make sure that no bubble was introduced in the system. SImilarly, the authors mixed 500 ml of seawater with porewater sediments to estimate the Rn concentration in porewater, they need to use radon free seawater ortherwise, they would need to determine the 226Ra activities from the water they used to to the mixing. 
  7. Section 2.5 the authors give 4 sinks but only three terms appear in the equation. It is not clear what is the difference between tidal output and mixing. 
  8. The authors often provide a range and an average. Instead of range from 17-136.7, I would write ranged from 17 to 140 or the range was 17-140. Also, I would be more careful with the significant figures. And finally, I think it is important that they provide a STD to the average because of the large range, we need to know if the two averages are significantly different or not. This needs to be applied throughout the entire paper. 
  9. In section 3.1, the authors speculate that high salinity is potentially due to domestic wastewater discharge. This is potentially a source of Rn that is not included in the mass balance. This source occurred during the sampling period at the HD location, which is also where the SGD estimates were the highest…. The authors need to provide evidence that this wastewater discharge did not affect the Rn mass balance. 
  10. Line 187, the authors said that the 226Ra activities did not change significantly in seawater during the measurement period, while figure 4 illustrates some changes with an increase of 226Ra that seems to be diurnal. Are these Ra226 peaks correlated with Rn increase, water depth, and salinity? 
  11. It is not clear when the authors provide a range of data that is negative : -97-140.9 Bq/m2h. Is it from -97 to -140.9 or from -97 to 140.9? and again, be careful with the significant figures. 
  12. One major comment on the paper is that the author estimated SGD fluxes at two locations but do not discuss the differences between these two locations. The discussion on seasonal SGD fluxes is very light and there is no discussion between the different sites. 
  13. Lines 238-240, this is an examples, but we can find this often in the paper: the authors present range of data but do not precise if these ranges are significantly different or not. Atomspheric loss: -5.68-8.18 Bq/m2h to compare with -7.09-14.69, and -3.72-4.95 … again, simplify with significant figures that need to be fixed throughout the paper, the average and the standard deviation are what is important to compare. For example line 259-260, we read 4744.74 Bq/m3 and 4787.81 Bq/m3, these numbers do not make sense, change to 4750 and 4790.
  14. I cannot find the values reported in table 1 in the text, which is problematic. the average values are not systematically reported in the text, and when it is, it does not match the value in the table; Additionally, there are 4 sink terms in the table and only three in the equations. 
  15. How are calculated F-tide-Out  and F-tide-Influx?  
  16. what means  54 +/- 117 Bq/m2h for HD dry season F-tide-outflux ??? The uncertainties in the table are often much larger than the value?? what does it mean? We can also read 13.5 +/- 135.7 ?? what does it represents?
  17. Regarding figure 7, how the authors explain the fact that the water depth  between TD dry and TD wet were very different, while the SGD fluxes were the same? Similarly, the SGD flux at HD dry is correlated with the water depth variation, while the TD dry SGD flux seems to be inversely correlated with the water depth variation? The discussion of the paper is relatively poor regarding the interpretation of the fluxes between the different sites. 
  18. section 4.. the last sentence is not clear “the SGD rates decreased or increased by 31% or 79% in the dry season, respectively” Does the SGD rates increase or decrease during the dry season?? change with appropriate significant figures. 5.1 +/- 2.3 10^2 Bq/m3 instead of 5215 +/- 2275 Bq/m3
  19. Table 2, range from ref 40, need to specify from which Ra isotopes the ranges are from or just write min - max for all 4 isotopes.  

Reviewer 3 Report

The manuscript was well written for publication in the target journal. Therefore, it can be published in the journal without any change.

Author Response

Response to Reviewer #3

Comments and Suggestions for Authors:The manuscript was well written for publication in the target journal. Therefore, it can be published in the journal without any change.

Response : Thank you very much for your positive comments and suggestions.

Round 2

Reviewer 2 Report

The authors have addressed all of my comments, I think the quality of the paper has improved and is now ready for publication.